# Indol-3-Carbinol and Quercetin Ameliorate Chronic DSS-Induced Colitis in C57BL/6 Mice by AhR-Mediated Anti-Inflammatory Mechanisms

**DOI:** 10.3390/ijerph18052262

**Published:** 2021-02-25

**Authors:** Sina Riemschneider, Maximilian Hoffmann, Ulla Slanina, Klaus Weber, Sunna Hauschildt, Jörg Lehmann

**Affiliations:** 1Department of Therapy Validation, Fraunhofer Institute for Cell Therapy and Immunology, 04103 Leipzig, Germany; sina.riemschneider@izi.fraunhofer.de (S.R.); M.Hoffmann@zytoservice-leipzig.de (M.H.); uschwerta@posteo.de (U.S.); 2AnaPath GmbH, CH-4410 Liestal, Switzerland; kweber@anapath.ch; 3Faculty of Life Sciences, University of Leipzig, 04103 Leipzig, Germany; shaus@server1.rz.uni-leipzig.de; 4Fraunhofer Cluster of Excellence Immune-Mediated Diseases CIMD, 04103 Leipzig, Germany

**Keywords:** aryl hydrocarbon receptor (AhR), chronic colitis, mouse model, indol-3-carbinol (I3C), quercetin, phytotherapy, inflammatory bowel disease (IBD)

## Abstract

Inflammatory bowel diseases (IBD), such as Crohn’s disease and ulcerative colitis, are multifactorial inflammatory disorders of the gastrointestinal tract, characterised by abdominal cramps, bloody diarrhoea, and anaemia. Standard therapies, including corticosteroids or biologicals, often induce severe side effects, or patients may develop resistance to those therapies. Thus, new therapeutic options for IBD are urgently needed. This study investigates the therapeutic efficacy and safety of two plant-derived ligands of the aryl hydrocarbon receptor (AhR), quercetin (Q), and indol-3-carbinol (I3C), using a translationally relevant mouse model of IBD. Q and I3C are administered by gavage to C57BL/6 wild-type or C57BL/6 *Ahr*^-/-^ mice suffering from chronic colitis, induced by dextran sulphate sodium (DSS). The course of the disease, intestinal histopathological changes, and in-situ immunological phenotype are scored over 25 days. Our results show that both Q and I3C improved significantly clinical symptoms in moderate DSS colitis, which coincides with a significantly reduced histopathological score. Even in severe DSS colitis I3C, neither Q nor the therapy control 6-thioguanine (6-TG) can prevent a fatal outcome. Moreover, treatment with Q or I3C restored in part DSS-induced loss of epithelial integrity by induction of tight-junction proteins and reduced significantly gut inflammation, as demonstrated by colonoscopy, as well as by immunohistochemistry revealing lower numbers of neutrophils and macrophages. Moreover, the number of Th17 cells is significantly reduced, while the number of Treg cells is significantly increased by treatment with Q or I3C, as well as 6-TG. Q- or I3C-induced amelioration of colitis is not observed in *Ahr*^-/-^ mice suggesting the requirement of AhR ligation and signalling. Based on the results of this study, plant-derived non-toxic AhR agonists can be considered promising therapeutics in IBD therapy in humans. However, they may differ in terms of efficacy; therefore, it is indispensable to study the dose-response relationship of each individual AhR agonist also with regard to potential adverse effects, since they may also exert AhR-independent effects.

## 1. Introduction

Inflammatory bowel diseases (IBD), with the major clinical forms Crohn’s disease (CD) and ulcerative colitis (UC), are chronic relapsing-remitting inflammatory disorders of the gastrointestinal tract with frequent extraintestinal manifestations. Typical symptoms include abdominal pain, diarrhoea, bloody stool, anaemia and weight loss [1,2,3]. IBD are multifactorial diseases with increasing incidence and prevalence worldwide. The incidence of CD in Europe ranges from 0.5 to 10.6 cases per 100,000 person-years, while the estimates for UC range from 0.9 to 24.3 per 100,000 person-years. At the turn of the 21st century, inflammatory bowel disease has become a global disease with accelerating incidence in newly industrialised countries whose societies have become more westernised. Although the incidence is stabilising in western countries, the burden remains high as prevalence surpasses 0.3%. These data highlight the need for research into preventing inflammatory bowel disease and innovations in health-care systems to manage this complex and costly disease [4]. The aetiology is so far unknown, but it is most probably a combination of genetic predisposition and environmental factors that impairs the intestinal epithelial barrier and results in a dysregulated immune response to the intestinal microbiota [5,6,7,8,9].

Dextran sulphate sodium (DSS) is a water-soluble, negatively charged, sulphated polysaccharide that damages the epithelial monolayer lining the large intestine. This disseminates the pro-inflammatory intestinal contents (e.g., bacteria and their products) into the underlying tissue [10]. DSS-induced colitis in mice is the most widely used inducible animal model in IBD research for studying pathogenesis and testing new therapeutics [11,12,13]. Recently, we have developed a refined model of chronic DSS-induced colitis in BALB/c mice that reflects typical symptoms of human IBD without risky bodyweight loss, usually observed in other DSS colitis models [14].

It has been shown that the aryl hydrocarbon receptor (AhR) plays a crucial role as a regulator of inflammatory processes on barriers in the organism [15,16]. Several reports have shown that AhR activation may result in anti-inflammatory effects in vitro and in vivo during inflammatory processes and the immune response [17,18,19,20,21]. The AhR is a ligand-activated transcription factor, persisting in the cytosol in its inactive form. Upon ligand binding, the receptor undergoes a conformational alteration, translocates into the nucleus and heterodimerises with the AhR nuclear translocator (ARNT). The AhR/ARNT complex binds to xenobiotic response elements (XRE) in promoter regions of AhR target genes (e.g., *Ahrr*, encodes for the AhR repressor; *Cyp1a1*, encodes for the metabolising enzyme CYP1A1) [22,23]. Besides this well-known pathway, several studies discussed additional interactions of the AhR with immune-relevant transcription factors. Kimura et al. described the interaction of the AhR with STAT1 and NF-κB, thereby inhibiting the transcription of *Il6* [24]. Furthermore, the AhR was reported to form complexes with the NF-κB subunits RelA and RelB, causing gene expression regulated by κB sites [25,26]. Numerous endogenous, as well as toxic or non-toxic exogenous AhR ligands, are known. Representative endogenous ligands of AhR are derivatives of tryptophan, such as kynurenine and the photoproduct 6-formylindolo[3,2-b]carbazole (FICZ) [27]. Exogenous toxic ligands, such as benzo[a]pyrene (BaP) or 2,3,7,8-tetrachlorodibenzo-1,4-dioxin (TCDD), are predominantly formed during organic combustion. These pollutants or their metabolites show carcinogenic or toxic effects [28,29]. Exogenous non-toxic ligands are often food-borne, such as indole-3-carbinol (I3C) or indirubin [29]. I3C occurs in cruciferous vegetables and can be converted to 3,3′-diindolylmethane (DIM) or indolo[3,2-b]carbazole (ICZ) under low pH in the stomach. I3C and its derivatives are known for their anti-proliferative effect on cancer cells and inhibition of transcription factors, such as NF-κB or the oestrogen receptor (ER) [30,31]. In contrast, binding to AhR activates the gene expression of AhR-regulated genes [32,33]. Another plant-derived AhR ligand quercetin (Q) was reported to induce antagonistic [34,35], and inhibitory effects on the activity of CYP1A1 [36] and kinases [37,38]. Furthermore, Q mediated inhibitory effects on lipopolysaccharide (LPS)-induced NF-κB activation and toll-like receptor (TLR)-4 signalling [39,40].

Thus, both natural AhR ligands I3C and Q seems to regulate AhR activity in different ways and may have a different impact on AhR-driven immunoregulation in chronic inflammatory processes. Therefore, we have studied the potential therapeutic efficacy of these phytochemicals in our refined chronic DSS colitis model. Since there is serious hope that AhR agonists may represent a new class of therapeutics in chronic inflammatory diseases, such as IBD, there is a need for reliable and standardised in-vivo models for preclinical selection and safety testing of numerous natural or synthetic AhR agonists.

## 2. Materials and Methods

### 2.1. Animals

Female C57BL/6JRj wild-type (WT) mice were originally purchased from Janvier Labs (Saint-Berthevin Cedex, France). C57BL/6 *Ahr*^-/-^ mice were bred in-house and regularly backcrossed to WT C57BL/6JRj mice, while the *Ahr* deletion was confirmed by genotyping via PCR. C57BL/6JRj WT and C57BL/6 *Ahr*^-/-^ mice of the same genetic background were used at 9–12 weeks of age. Mice were housed as five animals per cage in a temperature- and light/dark cycle-controlled environment (23 °C, 12 h/12 h light/dark, 50% humidity). They had free access to pelleted standard rodent chow and water ad libitum. All experimental procedures were approved by the Saxonian State Animal Care and Use Committee (Landesdirektion Sachsen, Leipzig, Germany, TVV 52/15) and were carried out in accordance with the European Communities Council Directive (86/609/EEC) for the Care and Use of Laboratory Animals. All animals were maintained in the animal care facility of the Fraunhofer Institute for Cell Therapy and Immunology (Leipzig, Germany). For organ collection, all animals were sacrificed using flow-controlled carbon dioxide (1 L/min). All efforts were made to minimise the suffering of the animals.

### 2.2. Adaptation of the Chronic DSS Colitis BALB/c Model to C57BL/6 WT and Ahr^-/-^ Mice

In a previous study, we developed a refined chronic DSS-induced colitis model in BALB/c mice, representing more likely the situation in human patients than the commonly used acute DSS colitis model [14]. To investigate the therapeutic effects of AhR ligands we intended to use AhR-deficient mice available on the C57BL/6 background. Therefore, we adapted the refined chronic DSS colitis BALB/c model to C57BL/6 mice. Two different protocols were tested for C57BL/6 WT mice. The first protocol was the same as specified for BALB/c mice administering 2% DSS (MW 36,000–50,000 Da, Lot-No. M7191; MP Biomedicals, Santa Ana, CA, USA) in autoclaved drinking water (w/v) for seven days, followed by 10 days of 1% DSS and the second phase of 2% DSS for another seven days, while the second protocol included 1% DSS for seven days, 0.5% DSS for 10 days and again 1% DSS for another seven days. Since *Ahr*^-/-^ mice are more sensitive to DSS than WT mice, as known from own observations, as well as from previously published work [41], the optimum DSS concentration for C57BL/6 *Ahr*^-/-^ mice, in terms of inducing disease symptoms that are comparable to those in C57BL/6 WT mice, had to be determined. Analogous to the test setup for WT mice, two different protocols were also tested for *Ahr*^-/-^ mice: (1) 0.4% DSS for seven days, 0.2% DSS for 10 days and again 0.4% DSS for another seven days or (2) 0.2% DSS for seven days, 0.1% DSS for 10 days and again 0.2% DSS for another seven days. In all protocols, DSS solutions were exchanged every 3–5 days.

Animals were inspected daily for overall physical and behavioural appearance. Scores for bodyweight, stool consistency and colonic haemorrhage were assessed daily. For the bodyweight score, the percentage of bodyweight loss was calculated relative to the bodyweight at day 0, with 2% bodyweight loss corresponding to 0.4 score points but a maximum score of 5 points. Stool consistency was evaluated from score 0 for normal stool consistency—1 for soft but still formed, 2 for very soft up to score 3 for very soft and liquid stool. Additionally, score 4 was given when there was blood in the stool. The clinical score was calculated as the average of the scores for bodyweight loss, stool consistency, and colonic haemorrhage. The scoring system for stool consistency and colonic haemorrhage is shown in Appendix A.

### 2.3. Drug Treatment

Drug solutions or homogenous suspensions were prepared in PBS (0.15 M NaCl) containing 1% hydroxyethyl cellulose (Sigma Aldrich, Taufkirchen, Germany). The following test compounds and doses were used: 1 mg/kg 6-thioguanine (6-TG) (Santa Cruz Biotechnology, Dallas, TX, USA), 50 mg/kg quercetin (Q), or 20 mg/kg indol-3-carbinol (I3C) (both from Sigma Aldrich). Drugs were administered by oral gavage from days 1–10 and 18–22 in a volume of 10 ml/kg. Since only I3C (20 mg/kg) showed therapeutic efficacy in WT mice, this substance was used in the survival experiment with *Ahr*^-/-^ mice. Administration volumes were calculated daily based on the individual bodyweight. Treatment control animals received vehicle only. Since both healthy WT and *Ahr*^-/-^ mice did not show any signs of colitis, only healthy WT animals were used as the healthy control group, according to 3R principles.

### 2.4. Colonoscopy

Weekly, the distal colon was exemplarily analysed by colonoscopy in two mice per group using a small animal endoscope (Karl Storz Endoskope Berlin, Berlin, Germany), starting at day 8. Mice were anesthetised using isoflurane (Isofluran CP^®^, cp-pharma, Burgdorf, Germany). For comparability, colonoscopy was always performed with the same animals. Representative pictures were taken with the COLOVIEW^®^ System Mainz (Karl Storz).

### 2.5. Gut Permeability In Vivo

Permeability of the intestinal epithelial barrier was assessed using the fluorescein isothiocyanate (FITC)-dextran (4000 Da) method [42]. Briefly, three mice per group were given FITC dextran (Sigma Aldrich) orally at 600 mg/kg at days 7, 14 and 21. After 1 h, submandibular blood was taken, and serum was prepared. The recovery rate of FITC-dextran in the serum was measured using a spectrophotometer (excitation: 485 nm; emission: 530 nm; Infinite^®^ M1000 PRO, Tecan, Männedorf, Switzerland).

### 2.6. Dissection

On day 25, animals were sacrificed in deep anaesthesia using carbon dioxide. The abdomen was opened by a longitudinal incision. The colon was dissected, and the length from the caecum to anus was measured. Two parts of the most distal colon were taken for histological evaluation.

### 2.7. Histological Evaluation

For each mouse, two samples from the distal colon were fixed in 4% phosphate-buffered formaldehyde for 24 h, dehydrated in grade ethanol, and embedded in paraffin. Tissue sections were cut at 3 µm on a rotary microtome (RM2255; Leica, Nussloch, Germany), mounted on glass slides and dried on a hotplate (60 °C). Sections were cleared, hydrated, and stained with haematoxylin and eosin (H&E) using standard protocols. These sections were evaluated in a blinded fashion by an experienced pathologist with regard to fibrosis in mucosa and submucosa, infiltration of granulocytes, lymphocytes and macrophages, as well as extent of oedema and necrosis (Appendix A). The histopathological score was calculated as the mean grades for these parameters.

For immunofluorescent analysis, tissue sections were blocked with 1% foetal bovine serum in PBS for 15 min. Blocked samples were incubated with primary antibody (polyclonal rabbit anti-mouse FoxP3, # ab54501; abcam, Cambridge, UK; monoclonal mouse anti-mouse RORγt–clone: Q31-378, # 562663; BD Pharmingen, Heidelberg, Germany; monoclonal rat anti-mouse F4/80–clone: F4/80, in house production; polyclonal rabbit anti-mouse MPO # ab9535; abcam; polyclonal goat anti-mouse IgA, # SBA-1040-01; SouthernBiotech, Birmingham, AL, USA; polyclonal rabbit anti-mouse CLDN1, # LS-B6327; LSBio, Seattle, WA, USA) at a dilution of 1:50 to 1:500 at room temperature (RT) for 2 h or 4 °C overnight. After washing, samples were incubated with secondary antibody (goat anti-rabbit IgG, biotinylated, # A16100; Thermo Fisher Scientific, Waltham, USA; goat anti-mouse IgG, biotinylated, # 115-066-072; Jackson ImmunoResearch, Newmarket, UK; donkey anti-rat IgG, biotinylated, # 712-065-153; Jackson ImmunoResearch; rabbit anti-goat IgG, biotinylated, # 305-065-003; Jackson ImmunoResearch; goat anti-rabbit IgG, Cy3-conjugated, # 111-165-144; Jackson ImmunoResearch) at a dilution of 1:200 to 1:500 at RT for 1 h. If biotinylated secondary antibodies were used, slides were washed and incubated at RT for 2 h with Cy3-conjugated streptavidin (Extravidin^®^-Cy3, Sigma Aldrich). Stained sections were washed and mounted with DAPI-containing mounting medium (Fluoroshield™ with DAPI, Sigma Aldrich).

Pictures were taken and digitised using a full-slide scanner (AxioScan.Z1; Zeiss, Jena, Germany and Olympus UC30; Olympus, Hamburg, Germany). Quantification of immunofluorescent staining was done with ImageJ software (Version 1.46r; Wayne Rasband, National Institutes of Health, Bethesda, MD, USA). Signals for Cy3 and DAPI were measured in a previously defined region of interest (ROI), applying previously saved greyscale limits. Marker expression was presented as Cy3-positive area relative to the DAPI-positive area [%]. For each treatment group, two sections per animal were analysed.

### 2.8. Statistical Analysis

All data were analysed using Prism 6 for Windows (GraphPad Software Inc., La Jolla, CA, USA). Line charts for clinical parameters are presented as mean ± SD. Significant differences between these data sets were estimated by two-way analysis of variance (ANOVA).

Scatter plots for clinical parameters, colon length and histological analysis are presented as individual data points. Significant differences between these data sets were estimated by Kruskal-Wallis one-way ANOVA (if normally distributed) or by one-way ANOVA on Ranks (if the normality test failed). In the case of the normal distribution of data, the data sets were compared by the Holm-Sidak’s post-hoc test. If the normality test failed, the Dunnett’s post-hoc test was applied. Values were considered significantly different if *p* ≤ 0.05, with *p* < 0.01, *p* < 0.001 or *p* < 0.0001 denoting higher significance levels. Only comparison within the WT groups or within the *Ahr^-/-^* groups was performed. There was no statistical analysis performed between WT and *Ahr^-/-^*.

## 3. Results

### 3.1. Adaptation of the Chronic DSS Colitis BALB/c Model to C57BL/6 WT and AhR^-/-^ Mice

As a precondition for proving AhR dependency in terms of pharmacological activity of plant-derived AhR ligands the chronic DSS-induced colitis model in BALB/c mice, previously published by our group [14,43], was adapted to C57BL/6 WT and C57BL/6 *Ahr^-/-^* mice. The setup for the protocol adaptation of colitis induction is specified in Section 2.

In contrast to BALB/c mice, the protocol 2%/1%/2% DSS resulted in a severe course of colitis around day 12 in C57BL/6 WT mice. Thus, this strain revealed to be more sensitive to DSS than BALB/c mice (Figure 1). However, the second protocol tested (i.e., 1%/0.5%/1% DSS) led to a more moderate course of colitis in C57BL/6 WT mice with very similar clinical symptoms as observed in BALB/c mice following oral administration of 2%/1%/2% DSS. Moreover, also in C57BL/6 *Ahr^-/-^* mice, the colitis induction protocol with higher DSS concentrations (i.e., 0.4%/0.2%/0.4%) caused a more severe course of colitis than the protocol with lower DSS concentrations (i.e., 0.2%/0.1%/0.2%). Thus, for proving pharmacological activity of the plant-derived non-toxic AhR ligands Q and I3C, we applied two different protocols of chronic colitis causing alternatively a severe or a moderate course of colitis in the C57BL/6 WT or the C57BL/6 *Ahr^-/-^* mouse model resembling different outcomes of IBD in human patients (Figure 1).

### 3.2. I3C Prevented Severe Outcome of DSS Colitis in C57BL/6 Mice in an AhR-Dependent Manner

In C57BL/6 mice, higher DSS concentrations (i.e., 2%/1%/2%) that have been otherwise shown to be tolerated in BALB/c mice caused a severe course of colitis reaching a critical stage in some animals (moribund). Those animals have been sacrificed, due to animal welfare (Figure 2a). Noteworthy, the treatment with I3C but not Q protected mice from the severe outcome of colitis caused by high DSS concentrations. The standard treatment with 6-TG weakened the severe course of colitis only slightly; however, that result was not statistically significant (Figure 2a). Although higher DSS concentrations (i.e., 0.4%/0.2%/0.4%) led to a severe outcome of colitis in *Ahr^-/-^* mice as well, here, I3C failed to improve the symptoms indicating that the therapeutic effect of I3C was AhR-dependent (Figure 2b).

### 3.3. Q and I3C Improved the Moderate Course of Chronic DSS Colitis in C57BL/6 Mice in an AhR-Dependent Manner

Using the moderate colitis model described above, the potential pharmacological activity of Q or I3C was studied in comparison to 6-TG, a pharmacologically active metabolite of azathioprine. Since azathioprine represents a standard therapeutic in human IBD, we used 6-TG as a therapy control in our chronic colitis model. On day 4 of oral uptake of DSS, the mice began to show clinical symptoms, such as diarrhoea and colonic haemorrhage (Appendix A). After 10 days, a static phase with chronic establishment of symptoms ensued (Appendix A). Both tested compounds, Q and I3C, were capable of improving the cumulative clinical score (Figure 3a). The significant reduction of clinical symptoms, as shown for stool consistency and colonic haemorrhage almost throughout the entire course of the experiment, was similar to the effect of 6-TG (Figure 3b,c and Appendix A). While I3C showed the strongest effects between days 6 to 15, Q and 6-TG consistently ameliorated symptoms over the entire course of the experiment. Of note, the therapeutic effects of Q and I3C could not be observed in C57BL/6 *Ahr^-/-^* mice, indicating an AhR-dependent mode of action for both substances. There were no significant differences in bodyweight changes between treatment groups and vehicle control, demonstrating that therapeutic treatment had only negligible effects on the bodyweight (Appendix A).

To monitor the macroscopic changes and signs of inflammation in the distal part of the colon in vivo, a colonoscopy of two animals per group was performed weekly starting at day 8 (Figure 4a). In the vehicle control group, disease symptoms were already detected at day 8, in particular, epithelial alterations, bloody lesions, and abscesses could be observed. Bloody lesions mostly occurred in the acute phase of the disease, but were rarely observed in the later chronic phase. Compared to vehicle-treated animals, 6-TG, Q, and I3C ameliorated all macroscopic symptoms, indicating an attenuation of DSS-induced inflammation in the colon.

Gut epithelial permeability was measured at days 7, 14, and 21 as an indicator of inflammation-induced loss of epithelial integrity (Figure 4b). As indicated by the high recovery rate of FITC-Dextran in the blood of vehicle-treated mice, tissue inflammation induced a significant loss of epithelial integrity. In contrast, mice treated with the test substances Q or I3C or with the treatment control 6-TG revealed a reduced permeability of the intestinal tissue, corresponding to reduced tissue inflammation as observed by colonoscopy.

### 3.4. Q and I3C Reduced Histopathology in Chronic DSS Colitis in C57BL/6 Mice

Applying the moderate colitis model in C57BL/6 mice, overall histopathological changes in the colon were studied and evaluated using a histopathological score based on a blinded evaluation of H&E-stained colon cross-sections by a trained pathologist (Figure 5a,b). Mice receiving the vehicle control showed a significantly higher histopathological score than animals treated with 6-TG, Q, or I3C (Figure 5b). This effect was not observed in C57BL/6 *Ahr^-/-^* mice. Interestingly, the histopathological score of vehicle-treated *Ahr^-/-^* mice was lower than for vehicle-treated WT mice. After dissection, colon length was measured as a marker for inflammation-induced scarring and fibrosis of the colon tissue. All therapeutic treatments were capable of partly preventing disease-induced colon shortening to a similar extent (Figure 5c).

To study the local inflammatory state of the colon tissue in more detail, colon cross-sections were stained for selected immune cell markers and digitised using a full-slide scanner. Immunofluorescent signals were quantified by calculating the ratio of the respective target signal to the DAPI signal of cell nuclei (Figure 6). An early marker of DSS-induced colitis is the infiltration of neutrophilic granulocytes that are characterised by the production of myeloperoxidase (MPO). In vehicle-treated animals, MPO expression was significantly increased. Treatment with I3C and Q reduced MPO expression almost to the level observed in healthy mice and was even more effective than 6-TG treatment (Figure 6a). A similar effect was observed for macrophages expressing the marker F4/80, indicating a reduced infiltration of innate immune cells into the colon tissue (Figure 6b). Since the ratio of pro-inflammatory T helper 17 (Th17) and regulatory T cells (Treg) is a major determinant of inflammation, the respective transcription factors RORγt and FoxP3 were analysed in situ (Figure 6c,d,g). As deduced from the results for RORγt-stained colon sections, Th17 cells were absent in the colon tissue of healthy mice, while the number of Th17 cells was increased in DSS-induced colitis. Administration of both test substances, Q or I3C, but also of the treatment control 6-TG significantly reduced RORγt-positive cells (Figure 6c,g). In contrast to RORγt, expression of FoxP3 as a marker for Treg cells was decreased in mice with DSS-induced colitis (Figure 6d,g). All treatments were capable of restoring FoxP3 expression in situ to the level observed in healthy animals suggesting the induction of Treg cells that support recovery and homeostasis at the gut barrier. Immunoglobulin A (IgA) secreted by B cells and plasma cells in the gut play a crucial role in the immune function of mucous membranes. All tested drugs significantly prevented DSS-induced upregulation of IgA expression (Figure 6e). As a marker for tissue integrity, tight junction (TJ) proteins form the continuous intercellular barrier between epithelial cells, which is required to separate tissue spaces and regulate the selective movement of solutes across the epithelium. In the chronic DSS-induced colitis model, DSS caused disruption of the epithelium and thereby a loss of TJ proteins, e.g., Claudin-1 (CLDN1). Accordingly, CLDN1 expression was down-regulated in DSS-induced colitis (Figure 6f). Treatment with both test compounds, Q or I3C, or with the treatment control 6-TG, restored CLDN1 expression to the level observed in healthy animals, suggesting that all three substances contribute to the restoration of the gut epithelial barrier. All effects observed after administration of Q or I3C were not apparent in *Ahr^-/-^* mice, indicating that the therapeutic effects were AhR-dependent.

## 4. Discussion

Phytochemicals have recently come into focus as potential therapeutics in IBD [44,45]. Those reports give hope that certain phytopharmaceuticals may substitute or complement current standard therapies, due to their convincing efficacy in terms of reharmonisation of the gut barrier with moderate side effects compared to standard therapeutics, such as dexamethasone or TNF-α inhibitors.

In this study, Q and I3C, two well-described plant-derived AhR ligands, were tested in two variants, i.e., severe or moderate course, of the chronic DSS-induced colitis model in C57BL/6 mice that has been adapted from a mouse model of chronic DSS colitis in BALB/c mice previously described by our group [14,43]. These models reflect the most relevant symptoms of human IBD, i.e., diarrhoea and colonic haemorrhage. DSS concentrations previously optimised for inducing moderate colitis in BALB/c mice caused a moribund phenotype in C57BL/6 mice; therefore, we have lowered the DSS concentrations for C57BL/6 mice to obtain moderate disease courses comparable to the BALB/c colitis model. Thus, we have established and standardised both a severe and moderate colitis model in C57BL/6 mice. To prove the impact of AhR ligation and subsequent signalling in terms of potential pharmacological effects of the drug candidates to be tested, our chronic DSS colitis model was adapted to *Ahr*^-/-^ mice on C57BL/6 background. As known from other reports [41,46], *Ahr*^-/-^ mice were expected to be more sensitive to DSS than WT mice. Therefore, DSS concentration had to be specially adjusted for *Ahr*^-/-^ mice to induce comparable clinical symptoms like in C57BL/6 WT mice. Finally, as for WT mice, two different protocols for inducing a severe or a moderate course of colitis have been established for *Ahr*^-/-^ mice, too.

In the first experimental setting, the potential therapeutic capacities of Q and I3C were assessed in the severe DSS colitis model in comparison to 6-TG. Standard treatment with 6-TG ameliorated the severe course of colitis and the fatal outcome only moderately (not statistically significant), whereas I3C significantly improved the course and the outcome of the disease. The different pharmacological effects of 6-TG and I3C are probably due to different modes of action of both drugs. 6-TG predominantly inhibits cell proliferation by blocking DNA synthesis by incorporating the base analogue 6-thioguanine instead of guanine into nucleotides. Since proliferating immune cells represent targets for this pharmacological effect 6-TG is applied as an immunosuppressive drug in chronic inflammation such as chronic colitis. However, myelosuppression may induce anaemia and other side effects, such as veno-occlusive liver disease [47,48]. Inhibition of cell proliferation was also reported for I3C. It is well known that I3C mediates anti-proliferative effects and promotes cell cycle arrest of tumour cells via gene regulation of, for example, cyclin-dependent kinases or tumour suppressor proteins [31,49,50]. Furthermore, it was shown that I3C decreased the proliferation of myeloid cells, such as dendritic cells, during differentiation [51]. Accordingly, we have previously shown that non-toxic doses of the AhR ligand BaP inhibited the proliferation of myeloid progenitors, while differentiating into macrophages in an AhR-dependent manner [21]. I3C may not only attenuate colitis symptoms by decreasing the number of immune cells, but also by exerting anti-inflammatory effects on immune cells [51]. Noteworthy, the beneficial effect of I3C was absent in *Ahr*^-/-^ mice, indicating that therapeutic effects of I3C are mediated by AhR activation. Together, these data underline the impact of AhR as a crucial regulator of several immune functions, in particular on epithelial barriers, and thus, should be considered as a therapeutic target in inflammatory diseases.

In a second experimental setting, the potential therapeutic capacities of Q and I3C were assessed in the moderate DSS colitis model in comparison to 6-TG. Since DSS-induced pathology is most prominent in the distal colon [52], the analysis was focused on this part of the intestine. It was shown that Q and I3C were as pharmacologically effective as the standard treatment with 6-TG in this colitis model. Both phytochemicals, Q and I3C, ameliorated clinical symptoms, such as stool consistency and colonic haemorrhage. Furthermore, treatment with both substances significantly reduced colon shortening, the permeability of the epithelial barrier, and histopathology. Similar results were already obtained for I3C and its metabolite 3,3’-diindolylmethane (DIM), as well as Q in rodent models of acute colitis [39,53,54]. To prove whether these effects were AhR-dependent, *Ahr*^-/-^ mice were included in this study. Since DSS-caused clinical symptoms could not be alleviated through the administration of I3C or Q in *Ahr*^-/-^ mice, it is very likely that the therapeutic effects of both phytochemicals observed in WT mice were AhR-dependent. The fact that in *Ahr*^-/-^ mice compared to WT mice, lower DSS doses are sufficient for inducing similar clinical symptoms impressively demonstrates that the epithelial integrity, as well as the homeostasis between the intestinal microbiota and the local immune response of the host, are much more fragile without expression of functional AhR. It is very likely that bacteria passing the damaged gut epithelial barrier induce stronger pro-inflammatory responses in *Ahr*^-/-^ mice, due to the dominance of IFN-γ-, TNF-α-, and IL-17-producing cells, i.e., ILC1, Th1 and Th17 cells, vs the low abundance of IL-22-producing cells, i.e., ILC3 and Th22 cells, in the lamina propria [19,55,56]. Nevertheless, this approach was sufficient to show that natural, plant-derived AhR ligands can restore clinical symptoms and histopathology in the moderate (Q or I3C) or even in the severe colitis model (I3C) in an AhR-dependent manner. Hence, this approach delivers clear evidence for AhR-dependency of pharmacological effects of I3C and Q in a translationally relevant mouse model of IBD. That, in turn, offers a solid basis for developing a new class of drugs for IBD. The mouse model, together with the analytic endpoints introduced in this work, represents a suitable and reliable system for preclinical safety and efficacy evaluation of new natural or synthetic AhR ligands that can also be used under good laboratory practice (GLP) conditions. However, a limitation of the mouse model used in this study is the synchronous administration of DSS and drug candidate. Thus, candidates selected from the chronic DSS colitis model should be verified by an alternative animal model of IBD. In this approach, the drug administration should not be started before the disease symptoms have been fully established.

To gain insights into the inflammatory process in the colon and to study the influence of AhR activation on particular inflammatory parameters, tissue cross-sections were analysed for the expression of several immune cell markers, IgA, and the TJ protein CLDN1. Since neutrophilic granulocytes and macrophages represent the first line of defence after infection, we assessed the appearance of both types of immune cells in the colon tissue. The enzyme MPO, which is predominantly expressed by neutrophils, catalyses the reaction of chloride to hypochlorite that is needed for recognition and phagocytosis of apoptotic cells [57]. MPO is secreted as a product of the oxidative burst during inflammation and is therefore almost absent in the colon tissue of healthy mice. The DSS-induced disintegration of the epithelium and following intrusion of luminal antigens into the sub-epithelial space of the colon tissue causes migration of neutrophils to the sites of microbial invasion. Accordingly, we observed a high level of MPO expression in vehicle-treated mice. Both plant-derived AhR ligands, I3C and Q, diminished MPO expression in colitis mice to levels similar as found in healthy animals. However, treatment with 6-TG has only weakly reduced the amount of MPO, supporting the idea that neutrophils and macrophages are not targeted by 6-TG treatment. Although the reduction of MPO expression by Q and the I3C metabolite DIM has already been reported in another mouse model of DSS colitis [39,58], in the present work, we could demonstrate that these effects were AhR-dependent. As deduced from F4/80 expression analysis, treatment with 6-TG, I3C, and Q as well reduced the number of infiltrating macrophages to a similar extent. Q was shown to inhibit dimerisation of the toll-like receptor (TLR)-4, which may cause reduced activation and infiltration of macrophages [40]. For I3C, it was shown that its effects depend on AhR-activated gene regulation, resulting in a shift to an anti-inflammatory cytokine secretion profile [51]. Furthermore, I3C and Q could induce a functional shift of macrophages to an anti-inflammatory phenotype, which has already been shown in vitro by other groups [59]. With respect to the adaptive immune system, Kimura et al. demonstrated that AhR activation inhibited LPS-induced expression of IL-6 in macrophages [24]. In turn, IL-6 was shown to induce differentiation of Th17 cells and to inhibit Treg differentiation [60,61]. Since the balance of Th17 and Treg cells, which can be influenced by AhR activation, significantly determines homeostasis vs inflammatory processes in several tissues [62], the expression patterns of the transcription factors RORγt (Th17 cells) and FoxP3 (Treg cells) were analysed in situ in the colon of colitis vs healthy mice. Since we have not co-stained the CD4 molecule on the tissue sections in this approach, RORγt^+^ cells may not only represent Th17 cells, but also ILC3. For that reason, we consider both of these cell populations together as an inflammatory pattern. In healthy mice, Th17 cells and/or ILC3 were almost absent, whereas Treg cells were present, maintaining normal gut homeostasis. In mice suffering from DSS-induced colitis, the number of Th17 cells and/or ILC3 increased, and the number of Treg cells decreased significantly. This inflammation-induced unbalance between both cellular patterns (Th17/ILC3 vs Treg) could be restored by all therapeutic treatments in this study. Of note, I3C had the most suppressive effect on the Th17/ILC3 pattern. This supports the hypothesis that reduced IL-6 expression by macrophages after I3C treatment resulted in reduced differentiation of Th17 cells. Similar results have already been reported for the AhR ligand TCDD in DSS-induced colitis, as well as for the I3C metabolite DIM in oxazolone-induced colitis [63,64]. In addition to T cell activation, IgA secretion by plasma cells is an important line of defence at mucosal barriers. DSS-induced colitis provoked increased secretion of IgA in the colon tissue as compared to healthy mice. This effect was reversed by all treatments tested in this study, confirming that these therapies were capable of regulating the DSS-induced adaptive immune response. In line with this finding, reduced expression of the TJ protein CLDN1 after DSS injury was effectively restored by all treatments. Interestingly, CLDN1 expression was further reduced by Q treatment in *Ahr*^-/-^ mice. This might be due to other regulatory effects of Q, such as influencing the activity of kinases [37,38].

The initial mechanism of how the activated AhR negatively regulates inflammation is still unknown. Since neutrophils and macrophages are the first inflammatory cells that migrate to the inflamed or damaged tissue, it is conceivable that the anti-inflammatory effect of AhR activation in macrophages is the initial step. Suppressed secretion of pro-inflammatory cytokines may prevent a strong inflammatory response by the adaptive immune system. In particular, decreased secretion of IL-6 by macrophages may cause a reduced differentiation of Th17 cells. These effects would avoid unnecessary tissue damage by immune cells and would enable faster tissue healing.

Importantly, our chronic DSS colitis model is characterised by damage of the intestinal epithelium promoting the invasion of gut microbiota into inferior layers of the bowel tissue, which, in turn, initiates a strong immune reaction resulting in further tissue damage. This pathogenic process is very similar to that in IBD patients. Furthermore, the chronic DSS-induced colitis model used in this study much better reflects the situation in human IBD patients than the commonly used acute DSS colitis model regarding chronicity, the extent of bodyweight loss, and histopathology (14). However, the principle mechanisms of epithelial damage differ between the mouse model and IBD patients since the colitis chemically induced by DSS in mice cannot reflect the various disease-inducing or -promoting factors contributing to IBD in humans, such as genetic defects, a dysbalanced immune response, or environmental factors. Nevertheless, the model of chronic DSS-induced colitis offers a suitable opportunity for preclinical testing of promising drug candidates.

## 5. Conclusions

In summary, the results from this study deliver clear evidence that plant-derived non-toxic AhR agonists can be considered as promising therapeutics in IBD therapy in humans. However, they may differ in terms of efficacy, particularly in terms of severe disease courses; therefore, it is indispensable to study the dose-response relationship of each individual AhR agonist, also with regard to potential adverse effects, since they may exert AhR-independent effects as well. Importantly, both plant-derived AhR ligands used in the present study induced fewer side effects, i.e., drug-induced weight loss, than the standard therapeutic 6-TG (a derivative of azathioprine). The mode of action of Q or I3C in chronic colitis is most likely restoring epithelial integrity by induction of tight-junction proteins and restoring the homeostasis of the innate and adaptive intestinal immune system by down-regulation of neutrophil and macrophage activity and by shifting the Th17/Treg ratio in favour of the Treg subpopulation, respectively.

## Figures and Tables

**Figure 1 ijerph-18-02262-f001:**
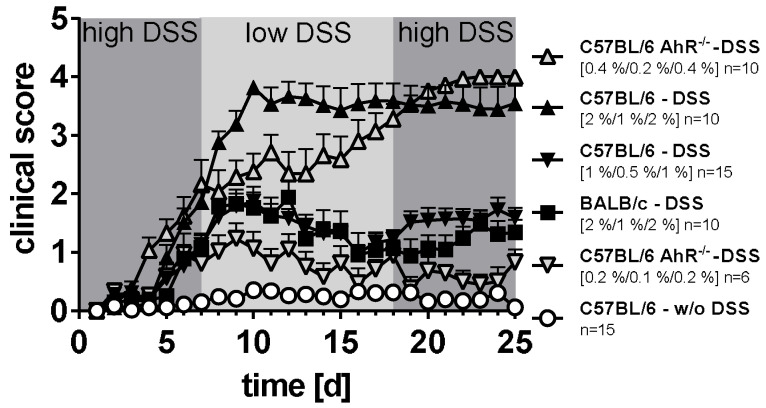
Courses of chronic DSS-induced colitis in BALB/c and C57BL/6 wild-type (WT) mice and C57BL/6 *Ahr* knock-out (*Ahr^-/-^*) mice. Mouse strains used in this study revealed unequal sensitivities to DSS. Therefore, the DSS concentration was adjusted individually for any of the three mouse strains to induce comparable symptoms as measured by the clinical score. Higher DSS concentrations, i.e., 2% or 0.4% DSS in drinking water for seven days, followed by 1% or 0.2% DSS for 10 days, and again 2% or 0.4% DSS for seven days, caused severe courses of colitis in C57BL/6 WT or *Ahr^-/-^* mice, respectively. Lower DSS concentrations, i.e., 1% or 0.2% DSS in drinking water for seven days, followed by 0.5% or 0.1% DSS for 10 days, and again 1% or 0.2% DSS for seven days, caused moderate courses of colitis in C57BL/6 WT or *Ahr^-/-^* mice, respectively. Control animals obtained water without DSS.

**Figure 2 ijerph-18-02262-f002:**
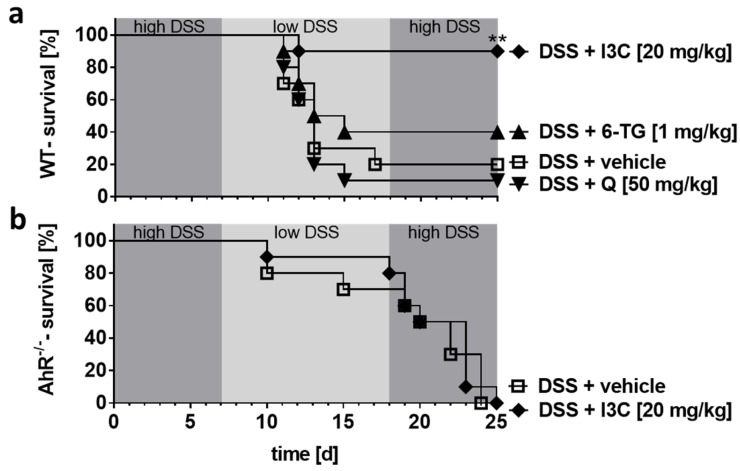
I3C diminished the fatal outcome of colitis induced by higher DSS concentrations in an AhR-dependent manner. Chronic colitis was induced in C57BL/6 wild-type (WT) (**a**) or *Ahr* knock-out (*Ahr^-/-^*) mice (**b**) by administering 2% or 0.4% DSS in drinking water for seven days, followed by 1% or 0.2% DSS for 10 days, and again 2% or 0.4% DSS for seven days, respectively. (**a**) WT animals were treated from days 1–10 and 18–22 with 6-thioguanine (6-TG; 1 mg/kg), quercetin (Q; 50 mg/kg), or indol-3-carbinol (I3C; 20 mg/kg) as indicated. While 6-TG and Q were not capable of preventing the fatal outcome of the disease induced by higher DSS concentrations, I3C reduced clinical symptoms even in the severe course of colitis, and thus, contributed to a significantly higher survival rate. (**b**) In *Ahr^-/-^* mice, I3C failed to prevent the fatal outcome of colitis. Control animals were administered with PBS (1% hydroxyethyl cellulose) by gavage (vehicle control). N = 10 WT or *Ahr^-/-^* mice per group; ** *p* < 0.01.

**Figure 3 ijerph-18-02262-f003:**
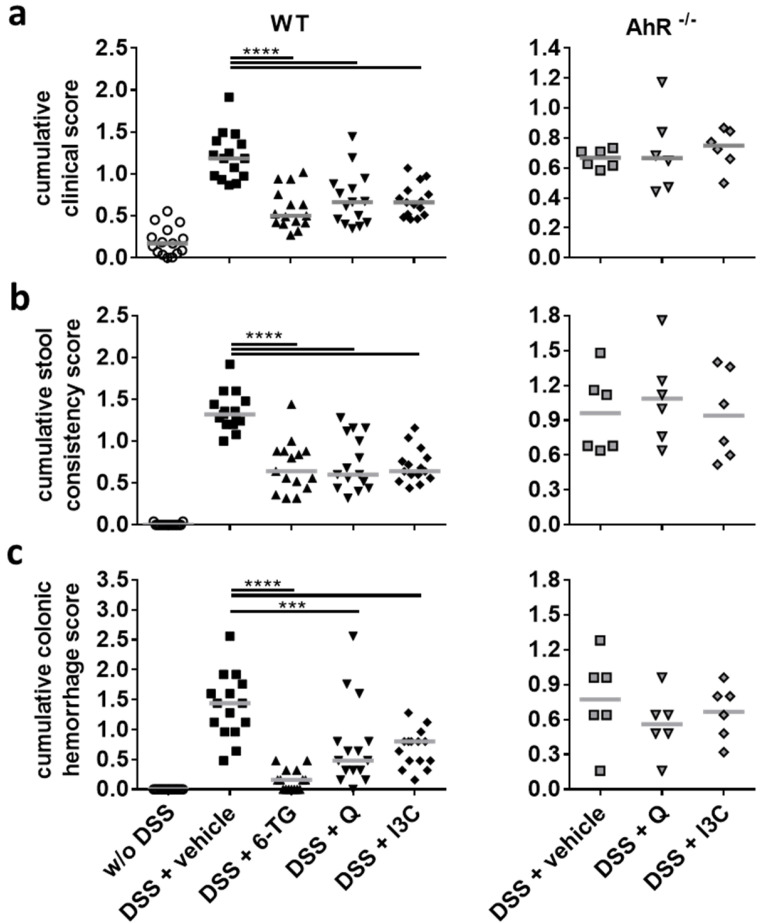
Q and I3C reduced clinical symptoms in the moderate course of chronic colitis in an AhR-dependent manner. Chronic colitis was induced in C57BL/6 wild-type (WT) or C57BL/6 *Ahr* knock-out (*Ahr^-/-^*) mice by administering 1% or 0.2% DSS in drinking water for seven days, followed by 0.5% or 0.1% DSS for 10 days, and again 1% or 0.2% DSS for seven days, respectively. Animals were treated from days 1–10 and 18–22 with 6-thioguanine (6-TG; 1 mg/kg), quercetin (Q; 50 mg/kg) or indol-3-carbinol (I3C; 20 mg/kg) as indicated. Control animals were administered with PBS (1% hydroxyethyl cellulose) by gavage (vehicle control). The clinical score was evaluated daily. Cumulative scores, which represent the average values out of all days of the experiment, for the clinical score (**a**) and its single parameters stool consistency (**b**) and colonic haemorrhage (**c**) are presented as individual data points with the median indicated. All cumulative scores could be significantly reduced by treatment with 6-TG, Q, or I3C. n = 15 WT mice or n = 6 *Ahr^-/-^* mice per group; *** *p* < 0.001 and **** *p* < 0.0001 vs the vehicle control group.

**Figure 4 ijerph-18-02262-f004:**
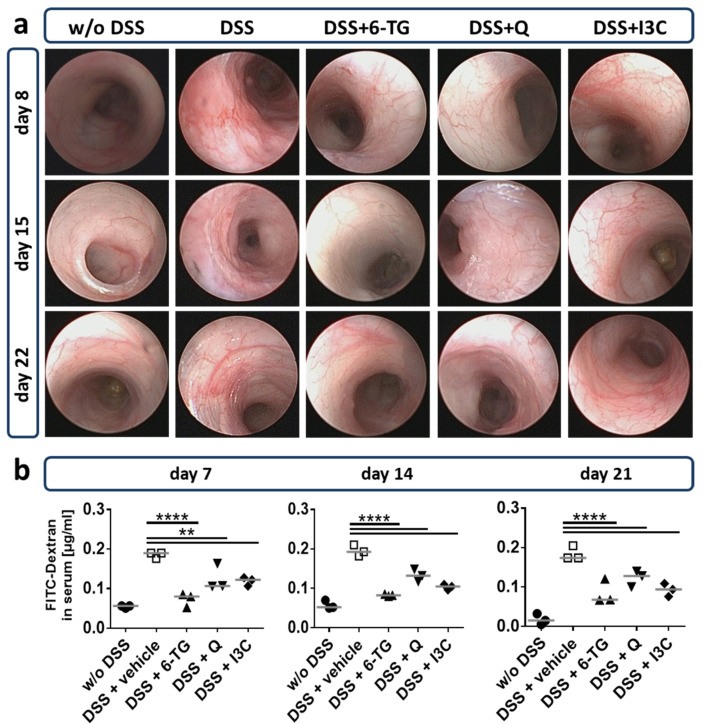
I3C and Q protected mice from DSS-induced intestinal lesions and gut permeability. Chronic colitis was induced in wild-type mice by administering 1% DSS in drinking water for seven days, followed by 0.5% DSS for 10 days, and again 1% DSS for seven days. Animals were treated from days 1–10 and 18–22 with 6-thioguanine (6-TG; 1 mg/kg), quercetin (Q; 50 mg/kg) or indol-3-carbinol (I3C; 20 mg/kg) as indicated. Control animals were administered with PBS (1% hydroxyethyl cellulose) by gavage (vehicle control). (**a**) On days 8, 15 and 22, representative pictures of the distal colon were taken by colonoscopy exemplarily from two mice per group. 6-TG, as well as Q or I3C, reduced visible lesions in the gut mucosa. (**b**) On days 7, 14 and 21, gut permeability was analysed using the FITC-dextran method and is shown as the concentration of FITC-dextran in the serum of individual mice with the median indicated. Both 6-TG and the AhR ligands Q or I3C significantly reduced the gut permeability caused by colitis. N = three mice per group; ** *p* < 0.01, **** *p* < 0.0001 vs the vehicle control group.

**Figure 5 ijerph-18-02262-f005:**
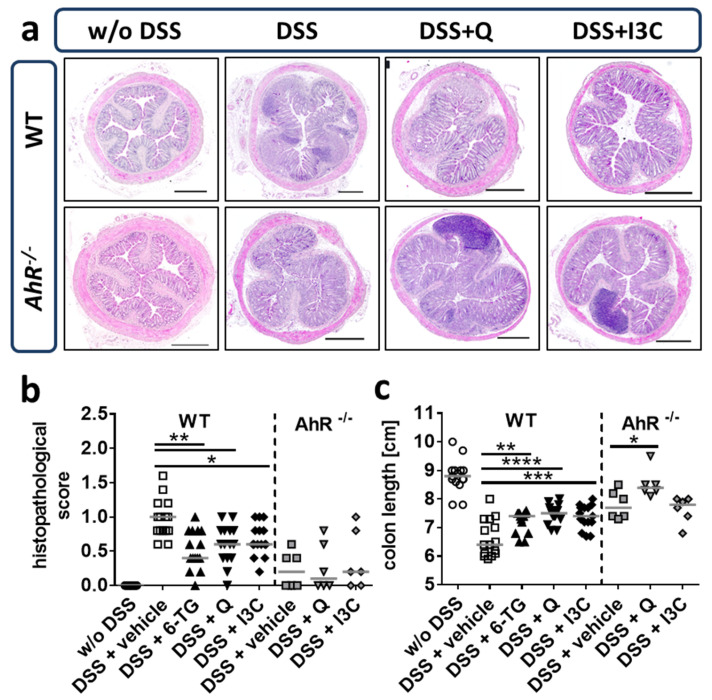
Q and I3C reduced histopathology in chronic DSS-induced colitis. Chronic colitis was induced in C57BL/6 wild-type (WT) or *Ahr* knock-out (*Ahr^-/-^*) mice by administering 1% or 0.2% DSS in drinking water for seven days, followed by 0.5% or 0.1% DSS for 10 days, and again 1% or 0.2% DSS for seven days, respectively. Animals were treated from days 1–10 and 18–22 with 6-thioguanine (6-TG; 1 mg/kg), quercetin (Q; 50 mg/kg) or indol-3-carbinol (I3C; 20 mg/kg) as indicated. Control animals were administered with PBS (1% hydroxyethyl cellulose) by gavage (vehicle control). (**a**) Exemplary photomicrographs of haematoxylin and eosin (H&E)-stained colon cross-sections from WT and *Ahr^-/-^* mice underwent chronic DSS colitis with or without treatment with AhR ligands (i.e., Q, I3C). H&E-stained cross-sections from healthy control mice (w/o DSS) are shown for comparison. Although the colon architecture of Q- or I3C-treated mice did not reach the normal state before administering DSS, restoration of the colon mucosa could clearly be achieved by the application of both AhR ligands. (**b**) The histopathological score was evaluated and is presented as individual data points with the median indicated. Similar to 6-TG, both Q and I3C significantly decreased the histopathological score in WT mice. Although the medians of all groups in *Ahr^-/-^* mice were relatively low, no differences between the treated and untreated groups could be observed. (**c**) Colon length was measured as a marker for DSS-induced inflammation and scarring and is presented as individual data points with the median indicated. In WT mice 6-TG, as well as Q and I3C, partly prevented shortening of the colon significantly. However, the amounts reached after treatment with all three drugs were still well below the median of healthy mice. The shortening of the colon in *Ahr^-/-^* mice was more moderate and could not be improved by I3C. Interestingly, the administration of Q led to weakly higher amounts. N = 15 WT mice; n = 6 *Ahr^-/-^* mice per group; * *p* < 0.05, ** *p* < 0.01, *** *p* < 0.001 and **** *p* < 0.0001 vs the vehicle control group. Scale bars = 500 µm.

**Figure 6 ijerph-18-02262-f006:**
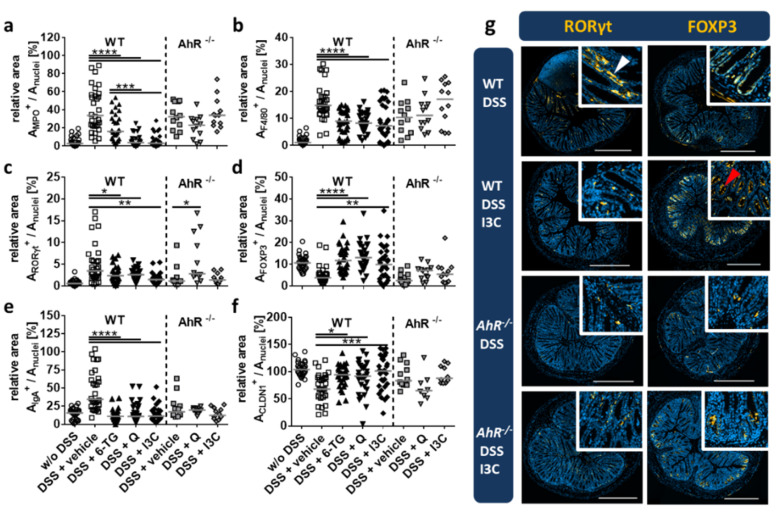
I3C and Q restored DSS colitis-associated histopathological changes of the bowel tissue. Chronic colitis was induced in C57BL/6 wild-type (WT) or *Ahr* knock-out (*Ahr^-/-^*) mice by administering 1% or 0.2% DSS in drinking water for seven days, followed by 0.5% or 0.1% DSS for 10 days, and again 1% or 0.2% DSS for seven days, respectively. Animals were treated from days 1–10 and 18–22 with 6-thioguanine (6-TG; 1 mg/kg), quercetin (Q; 50 mg/kg) or indol-3-carbinol (I3C; 20 mg/kg) as indicated. Control animals were administered with PBS (1% hydroxyethyl cellulose) by gavage (vehicle control). To prove the influence of the AhR ligands Q and I3C on the restoration of histopathological changes found in DSS colitis, relevant cellular markers, such as RORγt (**a**), FoxP3 (**b**), F4/80 (**c**), MPO (**d**), IgA (**e**), and CLDN1 (**f**), were quantified in colonic tissue by immunofluorescent staining and subsequent computer-assisted analysis using a full-slide scanner (AxioScan.Z1, Zeiss). In WT mice, most of the changes found in DSS colitis could be at least partly restored in animals treated with 6-TG, as well as Q or I3C. No such differences between the vehicle control and the Q- or I3C-treated groups were observed in *Ahr^-/-^* mice. N = 15 WT mice; n = 6 *Ahr^-/-^* mice per group; * *p* < 0.05, ** *p* < 0.01, *** *p* < 0.001 and **** *p* < 0.0001 vs the vehicle control group. Exemplary immunofluorescence images of selected groups for the markers FoxP3 and RORγt are shown (**g**). In the lamina propria of WT mice suffering from DSS colitis, a significant number of RORγt^+^ cells, which may represent Th17 cells or ILC3, was detected (left upper panel, white arrow). Under treatment with I3C, the number of RORγt^+^ cells in the lamina propria was significantly reduced, while the number FoxP3^+^ cells, which represent Treg cells, was significantly elevated (right upper panel, red arrow). Scale bars = 500 µm.

## Data Availability

All data generated or analysed during this study are included in this published article or are available as supporting material. The animal models established in this study are available for basic as well as applied research and for preclinical drug development (GLP level).

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
