# Peer review of "Indol-3-Carbinol and Quercetin Ameliorate Chronic DSS-Induced Colitis in C57BL/6 Mice by AhR-Mediated Anti-Inflammatory Mechanisms"

_ijerph, 2021, doi:10.3390/ijerph18052262_

Round 1
Reviewer 1 Report
In the following I am going to list my critics and suggestions towards the manuscript:
- According to the material and method section, the animals were treated from day 1 -10 and day 18 – 22. As no further indicated, I assume that the days refer to the treatment of the animals with DSS. I would like to ask the authors, on what basis this course of treatment was designed and why a break of 8 days was included.
- The authors showed that AhR-deficient mice are more sensitive towards DSS-treatment compared with wildtype mice. Therefore, different DSS-concentrations were used to induce a comparable severity of induced colitis. For both the mild and the severe course much lower DSS-concentrations were needed to induce a comparable clinical score in AhR-deficient mice. Thereby, Figure 1 reveals two different issues. Even if the severe outcome is similar in terms of the clinical score, the shape of the curves of wildtype and AhR-deficient mice varies tremendously. In case of the mild course of DSS colitis, the clinical score is almost two times lower in AhR-deficient mice compared to the proficient control.Due to this, it is rather questionable whether these strains can be compared in regard to their response to AhR-agonist treatment.
- In supplementary figure S1 the percentage change of the body weight is depicted. Is there an explanation why all groups, even the vehicle treated group, show a drastic reduction of the body weight around day 10? I would expect mice with 9 – 12 weeks of age to gain weight over the time course of the experiment, at least for the vehicle treated group.
- Figure 2 depicts the survivability of DSS-treated mice with a severe course of disease (high DSS-concentration). In wildtype mice, only indole-3-carbinol was able to diminish the fatal outcome, whereas quercetin had no effect compared to the vehicle treated control group. Unfortunately, the authors decided to treat AhR-deficient mice only with indole-3-carbinol and not with quercetin. Taking account of these results, the effect of indole-3-carbinol on fatal outcome is presumably not AhR-related.
- Concerning the different effects on fatal outcome, I recommend to perform IHC stainings of the AhR in colonic tissue. Analyzing the nuclear translocation may give insights whether the AhR is activated to similar amounts upon indole-3-carbinol or quercetin treatment.
- Figure 5 shows that the treatment with quercetin and indole-3-carbinol results in a reduced histopathology in chronic DSS-induced colitis. For this, a histopathological score of HE-stained colon-cross-section was obtained. This analysis revealed a reduced score upon 6-thioguanine as well as quercetin and indole-3-carbinol treatment compared to the control treated wildtype mice. To perform the same statistical analysis, it would be necessary to include the control treated AhR-deficient mice in the bar graph.
- The authors claim an AhR-dependent effect of quercetin and indole-3-carbinol on DSS-induced colitis. In general, the used DSS-concentrations for the AhR-deficient mice showed a very weakly pronounced colitis. 6-thioguanince should also be included for AhR-deficient mice to analyze whether a significant improvement of symptoms can be achieved.
- For figure 6 I would like to endorse the authors to provide representative pictures of the IHC stainings.
Based on the given results, the authors suggest the AhR as a potential therapeutic target to treat the inflammatory intestinal disease colitis. Activation of the AhR by the plant-derived ligands quercetin and indole-3-carbinol improves symptoms of colitis and therefore these candidates might be considered as new therapeutic candidates. In the last point I mostly disagree with the authors. Per definition, pronounced symptoms needs to be present prior therapy. In contrast to this, the authors started with the treatment simultaneously with the chemical induction of colitis. Consequently, the given results rather give insights towards the role of the AhR in the onset and progression of colitis.
Reviewer 2 Report
Journal: IJERPH (ISSN 1660-4601)
Manuscript ID: ijerph-1080348
Type: Article
Title: Indol-3-carbinol and quercetin ameliorate chronic DSS-induced colitis in C57BL/6 mice by AhR-mediated anti-inflammatory mechanisms.
Authors: Sina Riemschneider , Maximilian Hoffmann , Ulla Slanina , Klaus Weber , Sunna Hauschildt , Jörg Lehmann *
The manuscript investigates novel therapeutic options for IBD and studies the therapeutic efficacy of two plant-derived ligands Q and I3C. The result indicates that these plant-derived ligands may be a promising therapeutic target for the treatment of IBD in humans. The manuscript lies within the scope of the journal, and it gives sufficient background about the research. The abstract clearly describes the aim of the study and provides information about the method being used. It states the key results clearly and succinctly. The introduction provides an overview of IBD and the challenges that are being faced in its treatment and why there is an urgent need to develop new treatments for IBD. The authors cite previous studies and work done in the field and clearly state the objectives of the study. The material and method section is well defined with all the required information. The results are divided into subsections based on different studies which makes it easy to follow and understand. The figures are clear, and the figure legends are well labeled with all the necessary information. The references follow all the guidelines. Overall, it is informative and novel research, and it is very straightforward and well presented; and helps in answering a few challenging questions about the treatment of IBD.
Comments:
- In the introduction section a little bit of statistics might be helpful. (for example: how many people worldwide suffer from IBD, it will emphasize as to why it is important to look at this disease.
- Were male and female both mice used in the study? If yes, were there any sex differences?
- What was the number of animals (n) used in each group?
- Was the colonoscopy performed on the same animals every week or animals were taken at random? If they are taken at random each week, how do you compare the results to each other?
- Why do you think the histopathological score of vehicle-treated Ahr-/- mice was lower than for vehicle-treated WT mice?
Author Response
Point-by-Point Reply
Reviewer # 2:
The manuscript investigates novel therapeutic options for IBD and studies the therapeutic efficacy of two plant-derived ligands Q and I3C. The result indicates that these plant-derived ligands may be a promising therapeutic target for the treatment of IBD in humans. The manuscript lies within the scope of the journal, and it gives sufficient background about the research. The abstract clearly describes the aim of the study and provides information about the method being used. It states the key results clearly and succinctly. The introduction provides an overview of IBD and the challenges that are being faced in its treatment and why there is an urgent need to develop new treatments for IBD. The authors cite previous studies and work done in the field and clearly state the objectives of the study. The material and method section is well defined with all the required information. The results are divided into subsections based on different studies which makes it easy to follow and understand. The figures are clear, and the figure legends are well labeled with all the necessary information. The references follow all the guidelines. Overall, it is informative and novel research, and it is very straightforward and well presented; and helps in answering a few challenging questions about the treatment of IBD.
Comments:
- In the introduction section a little bit of statistics might be helpful. (for example: how many people worldwide suffer from IBD, it will emphasize as to why it is important to look at this disease.
We like to thank the reviewer for this advice. In the revised version of the manuscript we have included a short statistical overview in the introduction section (page 2, lines 47-55).
- Were male and female both mice used in the study? If yes, were there any sex differences?
No, in this study only female mice were used.
- What was the number of animals (n) used in each group?
The number of animals (n) were indicated in each figure description separately.
- Was the colonoscopy performed on the same animals every week or animals were taken at random? If they are taken at random each week, how do you compare the results to each other?
The colonoscopy was performed in the same animals. Now, this has been specified in the material and methods section, paragraph 2.4 (page 4, lines 165-166).
- Why do you think the histopathological score of vehicle-treated Ahr-/- mice was lower than for vehicle-treated WT mice?
It was really difficult to find a DSS dose for Ahr-/- mice to induce colitis symptoms without a severe course. In agreement with 3R principles we used a dose shown to cause weak symptoms without the risk of a severe course with loss of mice. Therefore, the histopathological score of Ahr-/- mice with DSS-induced colitis was lower compared to WT mice fed with DSS in drinking water.
Leipzig, 31th January 2021
Dr. Jörg Lehmann

Round 2
Reviewer 1 Report
The authors respond to my critics and suggestions in a very detailed manner. In the following I am going to refer to every single point.
- The authors described clearly the chosen model and the necessity of the intermediate phase.
- In this study, wildtype and AHR-/- mice were treated with different DSS concentrations to induce the same extent of chronic colitis. In my point of view, this fact reveals that the chosen model is not optimal to analyze the effect of the AHR in induced colitis. Due to the lack of alternatives and in regard to the clinical score the selection of this model is comprehensible.
- My question was answered to my full satisfaction.
- I agree with the authors that the multitude of AHR ligands show various biological effects, which are not related to the activation of the AHR. AHR ligands are divided into agonists and antagonists, which mainly refers to their capability to induce the expression of specific genes, such as CYP1A1 or CYP1B1. In this study, the authors have chosen indole-3-carbinol (I3C) as an agonistic and quercitin (Q) as an antagonistic compound. The latter one is subject of controversial debate and can be assigned to the group of partial agonists. Members of this group show agonistic function when applied alone, but are able to abrogate the activation of the AHR by prototypic agonists, such as TCDD (HP Ciolino , P J Daschner, GC Yeh. Dietary flavonols quercetin and kaempferol are ligands of the aryl hydrocarbon receptor that affect CYP1A1 transcription differentially. Biochem J. 1999 Jun 15;340 (Pt 3)(Pt 3):715-22).
- I do not see why it is not possible to analyze the nuclear translocation of the AHR in already obtained paraffin sections. Especially in the light of point four it would be essential to unravel the mode of action of quercitin to draw clear conclusions from the experiments.
- Even if the effect of 6-thioguanine is not AHR related, the AHR is responsible for the expression of many drug metabolizing enzymes already at basal activity. Therefore, the activity of 6-thioguanine might differ in AHR-/- from the activity in wildtype mice.
- See point six.
- The authors included microscopic pictures for selected groups. Unfortunately, this image (Figure 6) is of very poor quality. In general, the quality of the implemented pictures dropped drastically and needs to be improved.
- I agree with the authors that animal models are crucial tools to validate new drug candidates. The mouse model in this manuscript resembles the established chronic form of IBD in humans and at this point I do not disagree with the authors. Nevertheless, the treatment started before a chronic state of inflammation could be established, as the mice were treated simultaneously with the application of DSS. Therefore, the conclusions concerning the therapeutic effect of AHR ligands on chronic colitis needs to be phrased more carefully.
Author Response
Please see the attached point-by-point reply.
